# Obesity Is Associated with Distal Migration of Pancreatic Adenocarcinoma to Body and Tail: A Multi-Center Study

**DOI:** 10.3390/cancers16020359

**Published:** 2024-01-14

**Authors:** Wisam Sbeit, Gil Gershovitz, Amir Shahin, Shhady Shhadeh, Mahmoud Salman, Maamoun Basheer, Tawfik Khoury

**Affiliations:** 1Gastroenterology Department, Galilee Medical Center, Nahariya 22100, Israel; wisams@gmc.gov.il (W.S.); gil.ggarshowitz@live.biu.ac.il (G.G.); amir.schahin@gmail.com (A.S.); sh7ady.it91@gmail.com (S.S.); maamon.basheer@mail.huji.ac.il (M.B.); 2Azrieli Faculty of Medicine, Bar-Ilan University, Safed 1311502, Israel; 3Department of Surgery, Shaare Zedek Medical Center, Jerusalem 91120, Israel; dr.mahmoud88@hotmail.com

**Keywords:** pancreas, pancreatic adenocarcinoma, distal migration

## Abstract

**Simple Summary:**

Pancreatic adenocarcinoma (PAC) is one of the most lethal types of cancer. Most cases of PAC occur in the head of the pancreas. This study aimed to identify predictors of non-head PAC. We showed that obesity represents a significant clinical predictor of distal PAC. To the best of our knowledge, this association has not been previously described in the English literature. We found a relatively high occurrence of distal PAC in our cohort, with a potential role of obesity in the development of distal PAC. We suggest proper intervention to possibly minimize the risk of the disease. Awareness of patients’ complaints regarding unexplained abdominal pain is crucial to avoid misdiagnosis of distal PAC, which tends to present at later clinical stages and may aid in improving outcomes.

**Abstract:**

(1) Background: Pancreatic adenocarcinoma (PAC) is one of the most lethal types of cancer. Most cases of PAC occur in the head of the pancreas. Given the proximity of the pancreatic head to the bile duct, most patients present clinically during early stages of the disease, while distally located PAC could have delayed clinical presentation. (2) Aims: To assess predictors of non-head PAC. (3) Methods: A retrospective multicenter study was conducted, including all patients who had endoscopic ultrasound (EUS) for pancreatic masses and who had histologic confirmation of PAC. (4) Results: Of the 151 patients included, 92 (60.9%) had pancreatic head cancer, and 59 (39.1%) had distal pancreatic cancer. PAC at body was the most common location in the distal PAC group (31 patients (52.5%)). Logistic regression analysis demonstrated a significant association of obesity with distal migration of PAC (OR 4.44, 95% CI 1.15–17.19, *p* = 0.03), while none of the other assessed parameters showed a significant association. Notably, abdominal pain was more significantly associated with distal PAC vs. head location (OR 2.85, 95% CI 1.32–6.16, *p* = 0.008). (5) Conclusions: Obesity shows a significant association as a clinical predictor of distal PAC. Further studies are needed to better explore this association.

## 1. Introduction

The global burden of pancreatic cancer has increased dramatically over the past few decades and is expected to continue to represent a leading cause of cancer-related mortality. The past two decades have seen a doubling in the global annual number of pancreatic cancers diagnosed. Pancreatic adenocarcinoma (PAC) is a lethal disease constituting the seventh highest cause of death from cancer in developed countries worldwide [1]. The incidence of PAC is continuously rising, as it is estimated to be the second leading cause of cancer-related mortality by the end of 2030 [2]. The etiopathogenesis of the disease is multifactorial and has an association with smoking [3] and family history [4]. In addition, the disease has been associated with physical inactivity, alcohol consumption and several medical conditions including chronic pancreatitis, diabetes mellitus and obesity [5]. Prognosis of PAC is extremely poor, with a typical 1-year survival rate of 24% and a 5-year survival rate of only 9% following disease diagnosis [6]. During the early stages of PAC, patients are usually asymptomatic [7], and the site of the malignancy is the leading factor determining clinical presentation with disease advancement [6]. About 85% of cases of PAC occur in the head of the pancreas, while 15% occur in the pancreatic body and tail [8]. Survival and prognoses differ between adenocarcinoma of the pancreatic head vs. adenocarcinoma of the body and tail due to later clinical presentation and, therefore, advanced stages of the disease in the body and tail vs. the head [9,10]. To date, there is controversy about whether the pancreatic cancer site affects patients’ prognosis and survival, as the data in this field have yielded conflicting results. However, a recent systematic review and meta-analysis demonstrated that patients with pancreatic head vs. body and tail adenocarcinoma had a 5% reduced risk of mortality [11]. Assuming that one of the main factors affecting disease prognosis is the location of the lesion in the pancreas [9,12], the identification of predicting factors, both clinical and demographic, of distal PAC locations while still asymptomatic may contribute to earlier diagnosis of the disease. Therefore, the aim of our study was to identify variables that may predict distal migration (body and tail) of PAC.

## 2. Methods

We performed an analytical observational retrospective multi-center study that included all patients diagnosed with PAC at two medical centers in Israel (Galilee Medical Center and Shaare Zedek Medical Center) from 2015–2020. Inclusion criteria were patients aged 18 years and above who had undergone endoscopic ultrasound (EUS) and were diagnosed with PAC, with histologic confirmation that was obtained by endoscopic-ultrasound fine needle biopsy. Exclusion criteria were patients diagnosed with non-adenocarcinoma pancreatic neoplasms, such as neuroendocrine neoplasms and metastatic cancer to the pancreas, a family history of cancers, or patients with a hereditary/genetic predisposition for PAC. The variables that were assessed were demographic data for each patient including age, sex and ethnic background, clinical parameters including body mass index (BMI), background diseases and clinical presentations, and data regarding the location of the tumor and prognosis. These data were obtained from the medical centers’ electronic medical records. The study protocol conformed to the ethical guidelines of the 1975 declaration of Helsinki and was approved by the human research committee (approval number NHR-21-0027). Written informed consent was waived by the ethics committees due to the retrospective non-interventional nature of the study.

### Statistical Analysis

Continuous variables with normal distribution were represented with informative statistics as arhythmical averages and standard deviation (±SD) or with median and interquartile range for variables that are not normally distributed. Categorical variables were represented in incidence and percentage tables. Several indices were collected, including age, gender, weight, ethnicity, obesity smoking, metabolic syndrome, alcohol consumption, BMI, and background diseases. Following data collection, a logistic regression model was constructed to anticipate distal migration (migration yes/no) according to the indices described. A minimal statistical significance of 5% was set, along with a statistical power of 80%. Statistical analyses were completed using the Statistical Package for Social Science (SPSS version 24.0, IBM, Chicago, IL, USA).

## 3. Results

### 3.1. Demographics and Baseline Characteristics

Of the continuous variables, only BMI was normally distributed according to the Kolmogorov–Smirnov test and was reported by mean ± SD; the other continuous parameters were not normally distributed and were thus reported by median ± IQR. Overall, 151 patients were included in the study. Of these, 92 patients (60.9%) had pancreatic head cancer (group A), and 59 patients (39.1%) had distal pancreatic cancer (group B). The most common location in group B was the pancreatic body in 31 patients (52.5%), followed by the pancreatic tail in 17 patients (28.8%) and the pancreatic neck in 11 patients (18.7%) (Figure 1).

The median ± IQR for age was 72.5 ± 19 years in group A vs. 69 ± 18 years in group B. The male/female ratio was almost similar in the two groups. Similarly, the distribution of Arab and Jewish patients was similar. Of note, there was no difference in the rate of background diseases between the two groups. The rates of smoking, metabolic syndrome, active alcohol drinkers, diabetes mellitus and hypertension were 28.3%, 1.1%, 6.5%, 50% and 58.7% in group A vs. 16.9%, 0, 5.1%, 61% and 66.1% in group B, respectively. Table 1 shows the demographics and baseline characteristics.

### 3.2. Clinical Presentations among the Study Cohort

The most common clinical presentation for the entire cohort was abdominal pain followed by weight loss. Between the two groups, abdominal pain was more prevalent in group B (81.4%) than in group A (59.8%). On the other hand, jaundice and pruritus were more prevalent in group A (52.2% and 9.8%) as compared to group B (1.7% and 0), respectively, while weight loss did not differ between the two groups. Notably, distant metastasis was found at the time of diagnosis in 31.5% of patients in group A as compared to 45.7% of patients in group B. Figure 2 shows the clinical presentations.

### 3.3. Logistic Regression Analysis to Identify Parameters Associated with Distal Migration of Pancreatic Cancer

Performing univariate logistic regression analysis of demographics and baseline parameters to investigate the association with distal migration of PAC, we found that only obesity was significantly associated with distal migration of PAC (OR 4.44, 95% CI 1.15–17.19, *p* = 0.03). We could not identify any other clinical parameter associated with distal migration, including age (OR 0.99, 95% CI 0.97–1.02, *p* = 0.58), male gender (OR 0.85, 95% CI 0.44–1.64, *p* = 0.63), smoking (OR 0.55, 95% CI 0.24–1.26, *p* = 0.16) and diabetes mellitus (OR 1.55, 95% CI 0.80–3.01, *p* = 0.19). Table 2 shows the logistic regression analysis of the demographics and baseline characteristics. With regard to clinical presentation, abdominal pain was significantly associated with distal PAC location (OR 2.85, 95% CI 1.32–6.16, *p* = 0.008). As can be expected, jaundice showed a lower association with distal location (OR 0.02, 95% CI 0.004–0.13, *p* < 0.0001). Moreover, there was no difference in the rate of distant metastasis upon presentation (OR 1.79, 95% CI 0.91–3.52, *p* = 0.09) (Table 3).

## 4. Discussion

Pancreatic tumors remain a leading cause of death worldwide, with pancreatic adenocarcinoma contributing to most cases of pancreatic malignancies [13,14]. An overall increase in incidence and incidence-based mortality due to PAC has recently been demonstrated [15], and it is expected that PAC will become the second most common cause of death from cancer by the year 2040 [16]. Surgical resection is the only chance of a cure from the disease; however, given the late disease presentation, only 15–20% of patients have a resectable disease upon presentation [17].

Previous studies describing the distribution of tumors within the pancreas have shown that the most common site of PAC is in the head of the pancreas (ranging between 67% to 84% of cases), followed by the distal parts (neck, body, and tail) of the pancreas (ranging between 13% to 17% of cases) [18,19,20]. This distribution has a great consequential impact on the presentation, management, outcome, and prognosis of the disease. Given the high lethality rate of PAC, our study aimed to assess predictors of distal migration of PAC that are believed to be associated with late clinical presentation, advanced disease, poor prognosis, and survival.

Our work also found that the pancreatic head is the most common location of PAC, but with a relatively different ratio compared with distal parts of the pancreas. We found a higher proportion of distal PAC (39.1%) compared to studies prevailing in the literature (up to 17%) [18,19,20]. Our findings of an increase in the rate of distal (body, tail) PAC is in line with a previous study reporting the Surveillance, Epidemiology, and End Results Program (SEER) registry (including 85,715 cases from 1973–2014), which showed a steady decrease in the proportion of head PAC from 75.1 to 63.7% and a consistent gradual increase in the rate of distal body/tail PAC from 24.9 to 36.3% [21].

Several studies have suggested that the anatomic location of pancreatic tumors represents a potential determinant of survival [22,23]. A previous systematic review and meta-analysis of a total of 93 studies including 254,429 patients showed that the long-term prognosis of head cancers was better than that of body/tail cancers (HR = 0.96, 95% CI: 0.92–0.99; *p* = 0.02) and that pancreatic head adenocarcinoma was an independent prognostic factor for increased survival [11].

This led us to investigate potential predictors of distal migration of PAC to the pancreatic body and tail as well as the precipitants of such events. Our study found that obesity, defined as BMI > 30, is associated with a statistically significant association with distal migration of PAC. The association between obesity and the development of several comorbidities, including cancer, has been extensively studied and proven and has been specifically shown to be a risk factor for PAC [24,25]. Investigation of the underlying mechanisms of such a correlation has been carried out and has shown that fat deposition in the pancreas (“fatty pancreas”) serves as a mediator between obesity and the development of pancreatic cancer, and several molecular pathways have been linked to the pathophysiology and carcinogenesis of this process [26,27,28]. The other mechanisms include: (1) insulin resistance and hyperinsulinemia, which is associated with amplified digestive enzyme protein translation, inducing local inflammation followed by recurrent and chronic pancreatitis, which is considered a major risk factor for pancreatic intra-epithelial neoplasia and PAC [29]; (2) gut microbiota. Recently, the complex effects of gut microbiota have been elucidated as being a major risk factor in the development and progression of PAC [30]. A previous prospective study revealed a decline in gut microbiome diversity and a specific microbial profile in Chinese patients with pancreatic cancer [31]. An imbalance in the distribution of gut microbiota, named intestinal dysbiosis, can lead to the initiation of chronic inflammation that is mediated by a bacterial translocation component, such as lipopolysaccharide, which activates Toll-like receptors, inducing innate immunity and enhancing cancer progression [32]. Additionally, lipopolysaccharide plays an important role in the initiation of a pro-inflammatory cascade by activating the nuclear factor-κappa B (NF-κB) pathway, leading to the production of pro-inflammatory cytokines [tumor necrosis factor alpha (TNF-α), interleukin (IL)-6, and IL-1] and leading to oxidative stress and an inflammatory state [33]; and (3) pancreatic microbiota, as a previous study demonstrated the presence of various bacterial taxa in the pancreatic tissue, and some of these bacteria were found to be inhabitants in the oral cavity [34]. Del Castillo et al. found a lower level of Lactobacillus bacteria and a relative abundance of oro-dental-related bacteria in patients with PAC as compared to non-cancer controls [34]. Moreover, Geller et al. found that pathogenic bacteria colonize the cancerous pancreatic tissue and are a component in the cancer microenvironment; they thus lead to a change in tumor course and might impact tumor sensitivity to chemotherapy [35].

To date, no one has attempted to elaborate on predictors for distal migration of PAC. Further studies are needed to exactly define the site of fatty infiltration within the pancreatic portions and to correlate it with the development of head or distal PAC.

The association between certain risk factors, such as smoking or diabetes mellitus, and the development of pancreatic cancer remains unclear. Several studies reported a link between smoking [3,36] and diabetes [37,38] and the development of PAC; however, these data are contradicted by other studies reporting no such link [39]. Our work revealed no significant association of smoking and diabetes with distal vs. head PAC.

To the best of our knowledge, no other published study in the English literature has focused specifically on the analysis of this association, and we believe that this association needs to be further explored, given the size of our study population, to determine if such a relation does in fact exist. Not surprisingly, clinical presentation of PAC may vary broadly depending on the location of the tumor. As expected, the results of our study showed that jaundice is more strongly associated with tumors in the pancreatic head. Previous studies have shown similar results, with pancreatic head cancer presenting at earlier clinical stages with clinical features of obstructive jaundice [18,40]. This can be explained by the mechanical obstruction of the common bile duct caused by the tumor, based on the size of the lesion and alteration of normal anatomy. Malignancies located more distally in the pancreas are less likely to inflict such anatomical consequences, and jaundice was therefore less common in these patients. However, in patients with distal PAC, abdominal pain was found to be more prominent in our study. This finding has been previously demonstrated, and the characterization of the pain has been described [10,41]. Therefore, our results may emphasize the importance of physicians’ awareness of complaints exhibited by patients regarding unexplained abdominal pain, especially in obese patients. In these cases, we advocate that one proceed with a radiological investigation of the pancreas, to avoid the possibility of misdiagnosing distal PAC. The appropriate timing for making a diagnosis may enable us to offer the appropriate treatment, which has important prognostic value because late diagnosis may lead to locally advanced or metastatic disease [11]. Therefore, we highly urge caregivers to remain aware of pancreatic adenocarcinoma and proceed with appropriate investigations in obese patients with complaints of abdominal pain.

Notably, in our study we found that distant metastasis was more common in distally located pancreatic adenocarcinoma (45.7%) as compared to pancreatic head adenocarcinoma (31.5%), with no statistical significance (*p* = 0.09). Distal pancreatic adenocarcinomas usually do not cause jaundice by obstructing the common bile duct, thus reducing the opportunity for early diagnosis when compared to adenocarcinoma of the pancreatic head. This difference may explain the worse prognosis of body and tail adenocarcinoma compared to that in the head location [31]. A previous study showed that pancreatic head adenocarcinoma had a significantly lower risk of distant metastasis as compared to pancreatic body and tail adenocarcinoma [42]. The relatively small sample size in our study probably yielded a higher rate of distant metastasis, but without statistical significance. Therefore, a larger cohort study is warranted to precisely assess the effect of the PAC location on the distant metastasis rate.

Potential limitations of our study are the size of the research population and its retrospective design. We believe a larger study population may yield improved results and aid in determining more risk factors associated with distal migration of PAC. Additionally, characterizing the exact pancreatic portions of fatty infiltration will enable us to assess two points: first, whether obesity causes diffuse pancreatic fatty infiltration, and second, whether the association of obesity with distal PAC is related to the infiltration of fat within the distal part of the pancreas.

## 5. Conclusions

Obesity shows a significant association as a clinical predictor of distal PAC. We found a relatively high occurrence of distal PAC in our cohort, with a potential role for obesity in the development of distal PAC. We suggest proper intervention to possibly minimize the risk of the disease. Awareness of patients’ complaints regarding unexplained abdominal pain is crucial to avoid misdiagnosis of distal PAC, which tends to present at later clinical stages, and this may aid in improving outcomes.

## Figures and Tables

**Figure 1 cancers-16-00359-f001:**
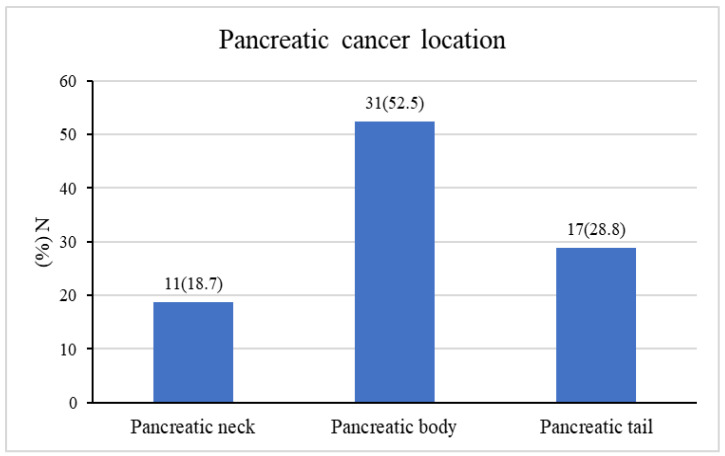
Distribution of non-head pancreatic cancer.

**Figure 2 cancers-16-00359-f002:**
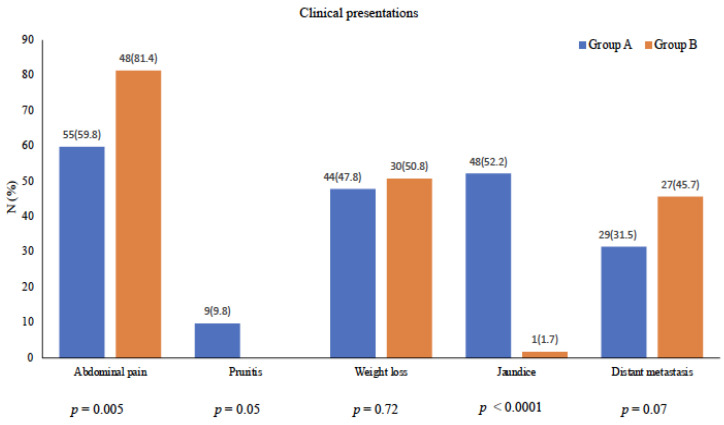
Clinical presentations of the study cohort.

**Table 1 cancers-16-00359-t001:** Demographics and baseline characteristics.

	Group A	Group B	*p* Value
Number	92	59	-
Age, median ± IQR (years)	72.5 ± 19	69 ± 18	0.29
Gender, N (%)			0.63
Male	52 (56.5)	31 (52.5)	
Female	40 (43.5)	28 (47.5)	
Ethnicity			0.95
Arab	30 (32.6)	19 (32.2)	
Jewish	62 (67.4)	40 (67.8)	
Weight, median ± IQR (kg)	68 ± 15	67 ± 21	0.14
Body mass index, mean ± SD	25.03 ± 3	24.9 ± 4.2	0.42
Obesity, N (%)	4 (4.3)	8 (13.6)	0.04
Smoking, N (%)	26 (28.3)	10 (16.9)	0.11
Metabolic syndrome, N (%)	1 (1.1)	0	0.75
Alcohol, N (%)	6 (6.5)	3 (5.1)	0.72
Overweight, N (%)	39 (42.4)	21 (35.6)	0.4
Diabetes mellitus, N (%)	46 (50)	36 (61)	0.18
Ischemic heart disease, N (%)	15 (16.3)	8 (13.6)	0.65
Hypertension, N (%)	54 (58.7)	39 (66.1)	0.36
Obstructive lung disease, N (%)	9 (9.8)	3 (5.1)	0.3
Renal failure, N (%)	6 (6.5)	0	0.17
Congestive heart disease, N (%)	8 (8.7)	4 (6.8)	0.67
Hyperlipidemia, N (%)	44 (47.8)	33 (55.9)	0.33

**Table 2 cancers-16-00359-t002:** Univariate logistic regression analysis of baseline parameters associated with distal migration.

	OR	95% CI	*p* Value
Age	0.99	0.97–1.02	0.58
Male gender	0.85	0.44–1.64	0.63
Ethnicity	1.01	0.50–2.04	0.97
Smoking	0.55	0.24–1.26	0.16
Metabolic syndrome	0.49	0.005–48.20	0.76
Alcohol	0.82	0.20–3.38	0.79
Overweight	0.68	0.34–1.39	0.29
Diabetes mellitus	1.55	0.80–3.01	0.19
Ischemic heart disease	0.82	0.33–2.08	0.68
Hypertension	1.36	0.69–2.68	0.37
Obstructive lung disease	0.54	0.14–2.04	0.37
Renal failure	0.11	0.005–2.54	0.17
Congestive heart failure	0.81	0.23–2.78	0.73
Hyperlipidemia	1.38	0.71–2.66	0.34
Obesity	4.44	1.15–17.19	0.03

**Table 3 cancers-16-00359-t003:** Logistic regression analysis of clinical presentations associated with distal migration.

	OR	95% CI	*p* Value
Abdominal pain	2.85	1.32–6.16	0.008
Weight loss	1.13	0.59–2.17	0.72
Jaundice	0.02	0.004–0.13	<0.0001
Pruritus	0.07	0.004–1.51	0.09
Distant metastasis	1.79	0.91–3.52	0.09

## Data Availability

Data are available from the authors upon reasonable request. The data are found at the Gastroenterology Department at Galilee Medical Center, Nahariya, Israel and will be available upon reasonable request.

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
