# Peer review of "Obesity Is Associated with Distal Migration of Pancreatic Adenocarcinoma to Body and Tail: A Multi-Center Study"

_cancers, 2024, doi:10.3390/cancers16020359_

Round 1

Reviewer 1 Report

Comments and Suggestions for Authors

This is a well written article with an interesting conclusion. But it seems that factors that predispose to the development of pancreatic cancer have not been examined such as family history of cancers or hereditary factors or gene variants. The existence of such factors may affect the results of the st;udy. If such patients are included they should be exluded from the analysis or if they are not this should be reported in the "Methods" section.

Author Response

This is a well written article with an interesting conclusion. But it seems that factors that predispose to the development of pancreatic cancer have not been examined such as family history of cancers or hereditary factors or gene variants. The existence of such factors may affect the results of the study. If such patients are included they should be excluded from the analysis or if they are not this should be reported in the "Methods" section.

Thank you. Those patients were excluded from the study. Data were added to the method section.

Reviewer 2 Report

Comments and Suggestions for Authors

Very interesting and innovative paper related to  the association of obesity, diabetes and cancer hot topic, namely with pancreatic adenocarcinoma. The manuscript is written clearly, concise and the retrospective analysis is ok.

Nevertheless, some aspects have to be improved:

1. Some spelling errors. For instance, the lack of space between words on line 42, as well as beginning the sentance with lowercase words. This aspect must be reviewed throughout the text.

2. On page 2, the paragraph that begins at line 103 is confusing, the % must be clearly presented.

3. Table 1 should have another column showing statistical significance (p). The prevalence of Metabolic syndrome on both groups is very low. Was the evaluation performed before or after treatment? Before or  after the patient looses more weight? Even if the % are correct, I think at least waist circumference analysis must be performed individually, as abdominal obesity has proved to be a RF for cancer independently of BMI.

4. Statistical significance should also appear on Figure 2.

5. The type univariate /multivariate of Logistic Regression Analysis is not specified on Table 2. 

6. Concerning the mechanisms that lead to neoplastic transformation of pancreatic cells, fatty infiltration is not the only mechanism. Insulin resistance and hyperinsulinism, chronic inflammation, pancreatic microbiota are some of the other mechanisms  involved. Exploring these aspects on discussion, could improve the manuscript interest.

7. At least, sub-analysing the influence of Diabetes therapy on PAC, namely  i-DPPIV and GLP1  agonists, if possible, could turn the manuscript even more interesting.

Comments on the Quality of English Language

I think English language is Ok, only spelling errors.

Author Response

Very interesting and innovative paper related to the association of obesity, diabetes and cancer hot topic, namely with pancreatic adenocarcinoma. The manuscript is written clearly, concise and the retrospective analysis is ok.

Nevertheless, some aspects have to be improved:

  1. Some spelling errors. For instance, the lack of space between words on line 42, as well as beginning the sentence with lowercase words. This aspect must be reviewed throughout the text.

Thank you. the text was reviewed and corrected.

  1. On page 2, the paragraph that begins at line 103 is confusing, the % must be clearly presented.

Thank you. corrected.

  1. Table 1 should have another column showing statistical significance (p). The prevalence of Metabolic syndrome on both groups is very low. Was the evaluation performed before or after treatment? Before or after the patient loses more weight? Even if the % are correct, I think at least waist circumference analysis must be performed individually, as abdominal obesity has proved to be a RF for cancer independently of BMI.

Thank you. The P value was added to table 1. Regarding the comment of the metabolic syndrome. The data was obtained from the patient file at the time of diagnosis. Unfortunately, due that our study is a retrospective study, we lack the data of waist circumference, as usually it is not assessed in our center. However, we agree that further study is needed to assess the rate of waist circumference among those patients, and indeed we will start plan such study. Thank you again for this important comment.

  1. Statistical significance should also appear on Figure 2.

Thank you. Added.

  1. The type univariate /multivariate of Logistic Regression Analysis is not specified on Table 2. 

Thank you. Added (univariate logistic regression analysis)

  1. Concerning the mechanisms that lead to neoplastic transformation of pancreatic cells, fatty infiltration is not the only mechanism. Insulin resistance and hyperinsulinism, chronic inflammation, pancreatic microbiota are some of the other mechanisms involved. Exploring these aspects on discussion, could improve the manuscript interest.

Thank you. Data were added in the discussion

  1. At least, sub-analysing the influence of Diabetes therapy on PAC, namely i-DPPIV and GLP1 agonists, if possible, could turn the manuscript even more interesting.

Thank you for your comment. Unfortunately, due to the retrospective study design we lack the data regarding the anti-diabetic medication. However, this will be our next project to assess whether those medication can impact the rate of pancreatic adenocarcinoma